# Impact of Bone Metastases on Patients with Renal Cell Carcinoma or Melanoma Treated with Combotherapy Ipilimumab Plus Nivolumab

**DOI:** 10.3390/biomedicines10112758

**Published:** 2022-10-31

**Authors:** Félix Pham, Samy Belkaid, Denis Maillet, Cyrille B. Confavreux, Stéphane Dalle, Julien Péron

**Affiliations:** 1Department of Dermatology, Immucare, Hôpital Lyon Sud, Hospices Civils de Lyon, 69310 Pierre-Bénite, France; 2Faculté de Médecine et de Maïeutique Lyon Sud, Université Claude Bernard Lyon 1, 69100 Lyon, France; 3Department of Oncology, Hôpital Lyon Sud, Hospices Civils de Lyon, 69310 Pierre-Bénite, France; 4Bone Metastases Expert Center CEMOS, Department of Rheumatology, Hôpital Lyon Sud, Hospices Civils de Lyon, 69310 Pierre-Bénite, France; 5Lyon-INSERM UMR 1033, Faculté Laennec, Université Claude Bernard Lyon 1, 69372 Lyon, France

**Keywords:** ipilimumab, nivolumab, renal cell carcinoma, melanoma, bone metastases, neutrophils-to-lymphocytes ratio

## Abstract

(1) Background: Ipilimumab plus nivolumab (combo-ICI) improves overall survival (OS) in patients with advanced renal cell carcinoma (RCC) or melanoma. The impact of bone metastases (BM) on survival outcomes of combo-ICI-treated patients is unknown. (2) Methods: This single-center retrospective observational study involved 36 combo-ICI-treated patients with advanced RCC and 35 with melanoma. Clinical and laboratory data preceding the initiation of combo-ICI were collected. Univariate and multivariate Cox proportional hazard models were used to assess the effect of BM on overall survival (OS) and progression-free survival (PFS). (3) Results: zNine RCC and 11 melanoma patients had baseline BM. In unadjusted analysis, baseline BM was associated with a poorer OS in the RCC cohort. Baseline BM did not have any impact on survival outcomes in melanoma patients. After adjustment on baseline performance status and on neutrophil-to-lymphocyte ratio (NLR), the impact of BM was no longer significant, but a NLR ≥ 3 was significantly associated with a poorer OS in the RCC cohort. (4) Conclusions: The presence of baseline BM seems to be associated with worse outcomes in RCC combo-ICI-treated patients, while its effect might not be independent from the inflammatory state (approximated by the NLR). BM seems to have no impact on the outcomes of melanoma combo-ICI-treated patients.

## 1. Introduction

The last decade, the combination of two immune checkpoint inhibitors (ICI), namely ipilimumab (anti-CTLA4) and nivolumab (anti-PD1), has dramatically improved patient outcomes in different malignancies including advanced melanoma and renal clear cell carcinoma (RCC): both conditions were the first for which these treatments obtained the approval of the European Medicines Agency. The use of this combination increased the progression-free survival (PFS) and overall survival (OS) of patients with advanced melanoma compared with ipilimumab alone; these results have been confirmed over a 5-year follow-up period [1,2]. Administering ipilimumab combined with nivolumab (combo-ICI) also improved the progression-free survival (PFS) and overall survival (OS) compared with sunitinib alone for patients with intermediate/poor-risk advanced RCC [3]. Furthermore, approximately one third of patients with advanced melanoma or RCC display long-term benefits from combo-ICI. However, these favorable outcomes are balanced by a significant toxicity burden and so far, no predictive biomarker is available to better stratify the benefits/risks ratio of the use of combo-ICI in advanced RCC and melanoma.

Bone metastases (BM) occur in about 30% of patients with advanced RCC [4]. In melanoma, BM occur at presentation in approximately one quarter of patients and are associated with poorer OS and melanoma-specific survival [5,6]. McKay et al. found that BM and liver metastases were associated with worse survival outcomes in advanced RCC patients treated with targeted therapy and the Meet-URO 15 study also reported an association between BM and poorer OS in advanced RCC patients receiving at least second-line nivolumab [7,8].

The impact of BM on the survival outcomes of patients treated with combo-ICI is unknown, although this information might help in the selection of the best regimen for patients with advanced RCC or melanoma. We, therefore, conducted a real-world retrospective analysis including patients from two French units to assess the prognostic impact of BM on patients with advanced RCC or melanoma treated with combo-ICI.

## 2. Materials and Methods

### 2.1. Setting and Participants

Patients included in this study were adults aged ≥18 years with advanced RCC or melanoma who initiated treatment consisting of at least one dose of combo-ICI administered intravenously in standard clinical practice or clinical trials at the Hôpital Lyon Sud, Hospices Civils de Lyon, France between July 2016 and January 2021. At the time of the study, the combo-ICI was not authorized in France for malignant pleural mesothelioma, non-small cell lung cancer or metastatic colorectal cancer; therefore, we did not include patients with these tumors. The study received the approval of the ethical review board of the Hospices Civils de Lyon (N° 21_244—22 June 2021) and Commission nationale de l’informatique et des libertés (CNIL, French data protection authority, N° 21_5244—22 June 2021). Patients did not receive any compensation for their participation.

### 2.2. Data Collection

Data collection was performed retrospectively from medical charts using a standardized data collection form by two dermatologists (F.P. and S.B.). Data collected included the following patient characteristics: age at treatment initiation, sex, history of autoimmune disease, Eastern Cooperative Oncology Group performance status (ECOG-PS), body mass index, smoking habits (active smoker, never smoker or former smoker since at least one year) before treatment with combo-ICI, type and grade of adverse events, red and white blood cell count, neutrophils-to-lymphocytes ratio (NLR) before the first dose of combo-ICI. An NLR threshold corresponding to the median value of the overall cohort was determined and used in regression models as a prognostic factor for survival outcomes. Disease characteristics were the following: tumor type (advanced melanoma or RCC), and number and localization of metastatic sites at treatment initiation. Treatment characteristics were the number of prior systemic treatments, prescribed dosage of combo-ICI, and prescription of a bone resorption inhibitor within 4 months after the finding of BM. The date of disease progression was determined by treating physicians according to standard practice. The progression had to be either clinical or radiological.

### 2.3. Statistical Analysis

The characteristics of patients in the two cancer groups were compared using two-tailed univariate analyses. A Fisher exact test was used to compare binary or qualitative variables. A Mann–Whitney test was used to compare quantitative variables.

The primary endpoint was OS. The secondary endpoints were PFS and extra-bone PFS. OS was defined as the delay between combo-ICI initiation (baseline) and death from any cause. PFS was defined as the delay between combo-ICI initiation and progression according to the physician assessment, or subsequent therapy or death from any cause, whichever occurred first. Extra-bone PFS was defined as the delay between combo-ICI initiation and extra-bone progression according to RECIST 1.1 criteria or death from any cause, whichever occurred first. When a new anticancer treatment was initiated as a consequence of a bone-disease progression, extra-bone PFS was censored at the time of the bone-disease progression. Survival probabilities were estimated using the Kaplan–Meier method and compared between groups using two-tailed log-rank tests. Covariates, including BM, were considered as statistically associated with PFS or OS if the associated *p* value was less than 0.05. To assess the adjusted effect of BM on OS and PFS, we used multivariate Cox proportional hazard models. All variables that had a statistically significant impact on OS or PFS in univariate analysis (using a threshold for *p* at 0.10), as well as the bone metastasis status (present or absent), were included in the multivariate models.

All analyses were performed using R statistical software (v4.1.1; R Core Team 2021, R Foundation for Statistical Computing, Vienna, Austria). Database follow-up was closed in February 2021. Few data were missing, and no data imputation was performed through the analyses.

## 3. Results

### 3.1. Patient and Treatment Baseline Characteristics

Between July 2016 and January 2021, 71 patients were included in this retrospective study. Among them, 36 (51%) patients were treated for an advanced RCC and 35 (49%) for a metastatic melanoma. Patients treated for an advanced RCC received 4 injections of 1 mg/kg ipilimumab (Bristol Myers Squibb, New York, NY, USA). Among the patients treated for a melanoma, 33/35 (94%) received 4 injections of 3 mg/kg ipilimumab, and the 2 remaining patients received 4 injections of 1 mg/kg ipilimumab. All patients received nivolumab (Bristol Myers Squibb, New York, NY, USA) in combination with ipilimumab, and in maintenance after the discontinuation of ipilimumab. The median [95% confidence interval, CI] follow-up duration after the initiation of combo-ICI was 22.1 [19.3; 30.2] months. A total of 9 (25%) patients with advanced RCC and 11 (31%) patients with melanoma had BM at presentation. In the overall cohort, patients with BM more frequently had an ECOG-PS ≥ 2 (7/20, 35% for BM patients and 3/51, 6% for non-BM patients; *p* = 0.0039). Patients with BM more frequently had liver metastases (11/20, 55% versus 12/51, 24%; *p* = 0.022; Table 1). Few patients with baseline BM received bone resorption inhibitor within four months after the discovery of BM (1/9 (11%) with RCC and 4/11 (36%) with melanoma.

### 3.2. Patient Outcomes according to the Presence of Bone Metastases

Within the overall cohort, patients with baseline BM receiving combo-ICI had a lower 12-month OS rate (52.4%, 95% CI: [33.8; 81.2]) compared with patients without initial BM (83.0%, [73.0; 94.6]; unadjusted hazard ratio [HR] = 2.5, [1.1; 5.8]; log-rank *p* = 0.027, Figure 1A). PFS was significantly shorter in patients with baseline BM (6-month PFS rate = 36.6%, [19.9; 67.2] for baseline-BM patients and 68.2%, [56.5; 82.4] for non-baseline-BM patients; unadjusted HR = 2.0, [1.0; 3.7]; log-rank *p* = 0.037, Figure 1B). After adjustment of NLR and baseline ECOG-PS, the impact of BM on OS (adjusted HR = 1.8, [0.64; 5.2]; log-rank *p* = 0.26) and PFS (adjusted HR = 1.7, [0.79; 3.7]; log-rank *p* = 0.17; Table 2) was no longer statistically significant.

Advanced RCC patients with baseline BM receiving combo-ICI had a significantly lower 12-month OS rate (41.7%, [18.5; 94.0]) compared with patients without baseline BM (12-month OS = 82.7%, [68.4–100]; unadjusted HR = 3.6, [1.1; 11.5]; log-rank *p* = 0.021, Figure 1C). The 6-month PFS rate reached 14.8%, [2.6; 86.0] in patients with advanced RCC and BM compared with 77.1%, [62.5; 95.1] in patients without BM (unadjusted HR = 6.7, [2.4; 19.2]; log-rank *p* < 0.0001, Figure 1D). After adjustment of ECOG-PS and NLR, the impact of BM on OS was not statistically significant anymore (adjusted HR = 1.5, [0.27; 8.6], log-rank *p* = 0.62) but remained consistent on PFS (adjusted HR = 6.4, [1.5; 26.8], log-rank *p* = 0.011; Table 3). Median time to extra-bone progression in patients with BM was 5.2 months, [0.6; not reached (NR)] compared with 15.5 months, [8.3–NR] in patients without BM (HR = 16.5, [2.0; 21.1]; *p* = 0.0016, Figure 2A).

Melanoma patients with BM had a 12-month OS rate of 60.6%, [36.8; 99.8] compared with 83.3%, [69.7; 99.7] for patients without BM (HR = 1.9, [0.55; 6.5]; log-rank *p* = 0.31). Similarly, the presence of BM had no statistically significant impact on the PFS (HR = 1.0, [0.39; 2.6]; log-rank *p* = 0.97) or extra-bone PFS (HR = 0.85, [0.31; 2.32], *p* = 0.75, Figure 2B) of melanoma patients.

### 3.3. Prognostic Factors of Effectiveness Other Than Bone Metastases

A NLR ≥ 3 was associated with shorter OS in patients with advanced RCC (adjusted HR = 16.7, [1.8; 156.6]; log-rank *p* = 0.014; Table 3) and its effect was independent of other prognostic covariates (BM and ECOG-PS). This was not observed in the melanoma cohort. Neither lung, central nervous system, nor liver metastases were significantly associated with poorer OS or PFS in the overall advanced RCC and melanoma cohort.

### 3.4. Immune-Related Adverse Events

Immune-related adverse events are summarized in Table 4. They were numerically more frequent and more severe in the melanoma cohort because of the higher dosage of ipilimumab. Skin-related adverse events such as rash and vitiligo were also far more frequent in the melanoma cohort.

## 4. Discussion

We found that baseline BM was not independently associated with shorter OS in advanced RCC patients. Rather, a high baseline NLR was predictive of worse OS in RCC.

Baseline BM were found in similar proportion than in previous studies [4,5,6]. In the melanoma cohort, there was clearly no association between BM and poorer survival outcomes. Results for RCC were not similar to those of the MM cohort. Indeed, in RCC combo-ICI-treated patients, we found initially, in unadjusted results, an association between shorter OS/PFS and BM at baseline. The potential negative impact of BM in advanced RCC seemed to be in line with previous studies of advanced RCC treated with nivolumab or targeted therapies [7,8]. This is an important point which raises the issue of the optimal management of BM in this population [9,10]. Notably, we observed that in our real-life cohorts, the prescription level of bone resorption inhibitors was low. This is a major issue since BM are responsible for severe pain and skeletal-related complications such as pathological fractures and medullary compressions leading to emergency surgeries, loss of autonomy, altered quality of life, oncological treatment delays and high healthcare costs [11,12]. In our BM-RCC cohort, this low prescription level is a bias that may be associated with poorer survival, independently of the efficacy of combo-ICI on BM. Furthermore, with the current available data on the absence of a significant difference between BM and non-BM, no specific restriction in terms of combo-ICI prescription should be applied in BM patients. On the contrary, these patients should additionally benefit from a modern dedicated approach to BM to optimize fracture risk evaluation, care, and skeletal event prevention [9,10,13].

Regarding the hypothesis that BM may weaken the effects of immunotherapy, some clues are available in the literature. First, the interaction between the receptor activator of NF-kB (RANK) and its ligand, RANKL, is dysregulated in BM-induced bone destruction [14]. For instance, RANKL has been found to be produced by regulatory T-cells in the tumoral microenvironment, and its interaction with RANK has been associated with an immunosuppressive environment promoting metastasis in mammary cancer in mouse models [15]. There could be potential therapeutic consequences with the development of bone-specific therapies to enhance the anti-tumoral response such as denosumab, an anti-RANKL monoclonal antibody, which may block tumoral progression as suggested by pre-clinical and clinical data [16,17,18]. To our knowledge, two ongoing clinical trials are prospectively investigating the efficacy of denosumab combined with pembrolizumab in advanced RCC (KEYPAD trial/NCT03280667) and with nivolumab or combo-ICI in advanced melanoma (CHARLI trial/NCT03161756).

In our study, NLR has appeared as an important factor associated with survival outcomes. Indeed, the association between baseline BM and OS in advanced RCC was not statistically significant after adjustment on the NLR and ECOG-PS. NLR is the neutrophils-to-lymphocytes ratio: it comes from a resuscitation score, and it has been successfully transferred in oncology settings initially in non-small cell lung carcinoma [19,20]. Indeed, an elevated NLR has been associated with shorter PFS and OS in RCC [21]. In our study, the adverse impact of BM on OS was neutralized when adjusted on a NLR ≥ 3. The causal relationship between BM and worse long-term outcomes is then uncertain. A first hypothesis to explain this observation might be that inflamed RCC are associated with more BM and worse ECOG-PS. Another explanation would be that BM in advanced RCC are associated with a more inflamed and immunosuppressive tumoral microenvironment, which in turn alters response to ICI as hypothesized above. In the Meet-URO 15 study, which evaluated prognostic factors in patients with metastatic RCC treated with nivolumab only, a NLR ≥ 3.2, baseline BM and intermediate or poor International Metastatic RCC Database Consortium score were all significantly associated with a worse OS in the multivariable model [8].

In melanoma, whereas a high NLR was associated with worse OS and PFS in two recent meta-analysis, this was not the case in our study [22,23]. It is worth noting that RCC is generally more associated with paraneoplastic syndromes than is melanoma, reflecting a probably more inflammatory state in RCC. Another consideration is that patients with BM in the study presented herein may have had a significantly worse ECOG-PS which is also a major prognostic factor in cancer patients.

Other predictive biomarkers have been proposed to better select patients who might benefit more from the combo-ICI or ICI in general than other treatments, but none is approved to date in melanoma or aRCC. For instance, the baseline PD-L1 expression level is used to prescribe pembrolizumab in metastatic non-small lung cancer but is not predictive of survival outcomes in melanoma treated with combo-ICI in the Checkmate-067 trial [2]. In aRCC, the PD-L1 expression level could not reliably distinguish patients who would drive more response or survival benefit from the combo-ICI in the Checkmate-214 trial [3]. A high tumor mutational burden is associated with better survival outcomes in melanoma patients treated with combo-ICI but has a low value in aRCC [24,25].

Surprisingly, patients with brain metastasis were not associated with poorer survival outcomes. Melanoma patients with brain metastases have always been the most challenging patients to treat, and this is particularly true for those with symptomatic brain metastases. The landmark trial Checkmate 204 has provided strong evidence of the intracranial activity and efficacy of ipilimumab plus nivolumab in melanoma patients with asymptomatic brain metastases [26]. However, patients with symptomatic brain metastases still have a very poor prognosis. In our study, even if we did not specifically study it, it is probable that most of the patients with brain metastases were asymptomatic, benefiting from the combo-ICI and explaining why brain metastases were not associated with poor survival outcomes [27].

There were several limitations in our study such as the small sample size resulting in a lack of statistical power owing to the monocentric nature of this study. Secondly, the retrospective nature of the study was inherently associated with residual confounding factors despite our effort to control it with a Cox model. iRECIST criteria were not considered in this study as they were not routinely used for all patients at the time they were treated with an ICI. We did not accurately describe the pattern of BM such as the number of skeletal localizations and their detailed topography. We also did not specifically investigate bone events such as fractures or whether the patients were discussed in bone metastatic multidisciplinary tumor boards. Therefore, it is critical not to extrapolate that combo-ICI might be less efficient in patients with baseline BM. The results of the present study are only exploratory, and no causal or definitive relationship can thus be inferred between baseline BM, elevated NLR and altered ECOG-PS in patients with RCC treated by combo-ICI.

To overcome the limitations of our work, future studies could prospectively study the impact of the NLR and better describe the presence, number, location, and treatment of BM in a more coordinated and multicentric fashion. Furthermore, subgroup analyses specifically studying patients with versus without BM and patients with “low” versus “high” NLR could be performed in prospective trials evaluating combo-ICI. Finally, a meta-analysis of every prospective clinical trial of combo-ICI would be able to better understand the potential prognostic and predictive effect of NLR and baseline BM.

## 5. Conclusions

Baseline BM in patients with advanced RCC and melanoma treated with combo-ICI were not significantly associated with worse OS, but our study certainly lacked statistical power. However, an elevated NLR was independently associated with poorer OS in patients with advanced RCC. Larger cohorts are needed to better understand the clinical and prognostic significance of BM and NLR in these populations.

## Figures and Tables

**Figure 1 biomedicines-10-02758-f001:**
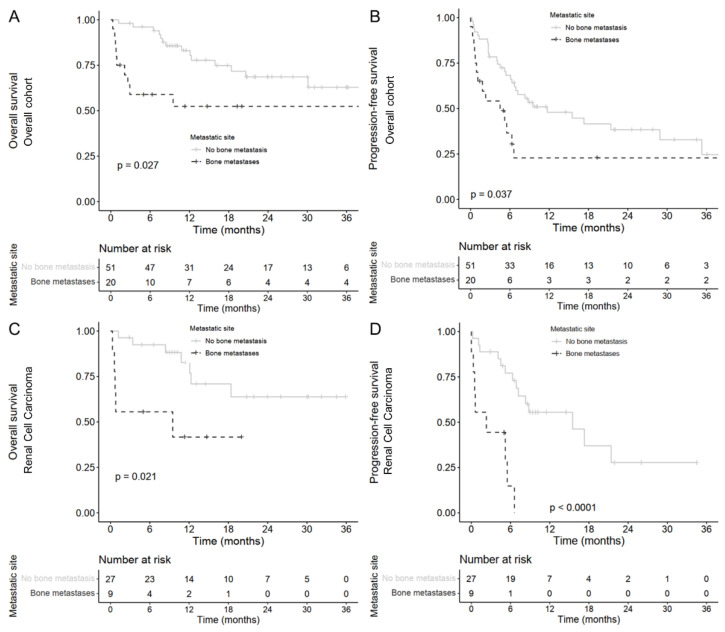
Kaplan–Meier estimates according to bone metastases for patients treated with ipilimumab plus nivolumab for (**A**) overall survival and (**B**) progression-free survival in the global cohort. Kaplan–Meier estimates according to bone metastases for patients treated with ipilimumab plus nivolumab for (**C**) overall survival and (**D**) progression-free survival in renal cell carcinoma.

**Figure 2 biomedicines-10-02758-f002:**
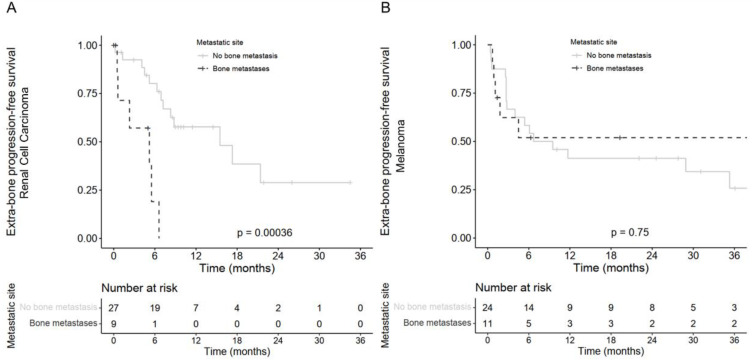
Kaplan–Meier estimates of the extra-bone progression-free survival according to bone metastases for patients treated with ipilimumab plus nivolumab (**A**) in patients with renal cell carcinoma and (**B**) in patients with melanoma.

**Table 1 biomedicines-10-02758-t001:** Baseline patients characteristics according to tumor type at initiation of ICI.

Variable	All Patients (*n* = 71)				Renal Cell Carcinoma (*n* = 36)				Melanoma (*n* = 35)			
	All Patients	BM (*n* = 20)	No BM (*n* = 51)	*p*	All Patients	BM (*n* = 9)	No BM (*n* = 27)	*p*	All Patients	BM (*n* = 11)	No BM (*n* = 24)	*p*
Age, years, median (25th–75th)	60 (52–69)	63 (48–69)	60 (53–70)	0.77	67 (56–71)	68 (54–71)	66 (57–72)	0.83	53 (45–67)	54 (46–69)	53 (45–66)	0.90
Gender male (%)	51 (72%)	12 (60%)	39 (76%)	0.24	28 (78%)	5 (56%)	23 (85%)	0.086	23 (66%)	7 (64%)	16 (67%)	1.0
PS ≥ 2 (%)	10 (14%)	7 (35%)	3 (6%)	0.0039	4 (11%)	3 (33%)	1 (4%)	0.041	6 (17%)	4 (36%)	2 (8%)	0.063
BMI (%)												
<18	0 (0%)	0 (0%)	0 (0%)	0.76	0 (0%)	0 (0%)	0 (0%)	1.0	0 (0%)	0 (0%)	0 (0%)	0.69
18–30	55 (77%)	15 (75%)	40 (78%)		28 (78%)	7 (78%)	21 (78%)		27 (77%)	8 (73%)	19 (79%)	
>30	16 (23%)	5 (25%)	11 (21%)		8 (22%)	2 (22%)	6 (22%)		8 (23%)	3 (27%)	5 (21%)	
≥3 metastatic sites	42 (59%)	17 (85%)	25 (49%)	0.0070	13 (36%)	6 (67%)	7 (26%)	0.046	29 (83%)	11 (100%)	18 (75%)	0.15
Type of metastases (%)												
Brain	19 (27%)	4 (20%)	15 (29%)	0.56	3 (8%)	1 (11%)	2 (7%)	1.0	16 (46%)	3 (27%)	13 (46%)	0.17
Lung	46 (65%)	14 (70%)	32 (63%)	0.78	27 (75%)	8 (89%)	19 (70%)	0.40	19 (54%)	6 (55%)	13 (54%)	1.0
Liver	23 (32%)	11 (55%)	12 (24%)	0.022	7 (19%)	3 (33%)	4 (15%)	0.33	16 (46%)	8 (73%)	8 (33%)	0.065
Bone	20 (28%)	20 (100%)	0 (0%)	NA	9 (25%)	9 (100%)	0 (0%)	NA	11 (31%)	11 (100%)	0 (0%)	NA
Active smokers (%)	16 (23%)	5 (22%)	11 (22%)	1.0	3 (8%)	0 (0%)	3 (11%)	0.56	13 (39%)	5 (45%)	8 (36%)	0.71
Any history of autoimmune disorder (%)	3 (4%)	1 (4%)	2 (4%)	1.0	1 (3%)	0 (0%)	1 (4%)	1.0	2 (6%)	1 (9%)	1 (4%)	1.0
NLR, NA = 4												
<2	15 (22%)	2 (11%)	13 (27%)	0.39	7 (22%)	1 (14%)	6 (24%)	0.24	8 (23%)	1 (9%)	7 (29%)	0.26
2–3	24 (36%)	7 (39%)	17 (35%)		11 (34%)	1 (14%)	10 (40%)		13 (37%)	6 (55%)	7 (29%)	
≥3	28 (42%)	9 (50%)	19 (39%)		14 (44%)	5 (71%)	9 (35%)		14 (40%)	4 (36%)	10 (42%)	
Previous treatment line (%)												
No	57 (80%)	14 (70%)	43 (84%)	0.30	30 (83%)	6 (67%)	24 (89%)	0.30	27 (77%)	8 (73%)	19 (79%)	1.0
Yes	14 (20%)	6 (30%)	8 (16%)		6 (17%)	3 (33%)	3 (11%)		8 (23%)	3 (27%)	5 (21%)	

BM: bone metastases; BMI: body mass index; ICI: immune checkpoint inhibitor; PS: performance status; NA: not available; NLR: neutrophils-to-lymphocytes ratio.

**Table 2 biomedicines-10-02758-t002:** Prognostic factors of overall survival and progression-free survival among the whole patients treated with ipilimumab plus nivolumab according to the Cox model.

Patient Characteristics	Cox Proportional Hazards Regression for Survival										
		Overall Survival					Progression-Free Survival				
		12 Months OS, % (95%CI)	Unadjusted Analysis		Adjusted Analysis		6 Months PFS, % (95% CI)	Unadjusted Analysis		Adjusted Analysis	
	N (%)		HR (95%CI)	*p*	HR (95%CI)	*p*		HR (95%CI)	*p*	HR (95%CI)	*p*
Age											
<70	56 (79%)	77.0 (66.3–89.4)	REF 1.2 (0.43–3.1)	0.77	NI	NI	59.9 (48.2–74.4)	REF 1.4 (0.72–2.8)	0.32	NI	NI
≥70	15 (21%)	65.0 (44.4–95.3)					59.3 (38.7–90.7)				
PS ≥ 2 (%)											
No	61 (86%)	78.3 (68.1–90.1)	REF 3.1 (1.2–7.8)	0.019	REF 1.6 (0.50–5.2)	0.43	64.6 (53.5–78.1)	REF 2.0 (0.91–4.3)	0.085	REF 1.4 (0.54–3.5)	0.50
Yes	10 (14%)	50.0 (26.9–92.9)					30.0 (11.6–77.3)				
≥3 metastatic site (%)											
No	29 (41%)	77.2 (62.6–95.4)	REF 0.91 (0.40–2.1)	0.82	NI	NI	57.2 (41.4–79.0)	REF 0.79 (0.43–1.5)	0.45	NI	NI
Yes	42 (59%)	72.2 (59.5–87.7)					61.2 (47.9–78.1)				
Bone metastases (%)											
No	51 (72%)	83.0 (73.0–94.6)	REF 2.5 (1.1–5.8)	0.027	REF 1.8 (0.64–5.2)	0.26	68.2 (56.5–82.4)	REF 2.0 (1.0–3.7)	0.037	REF 1.7 (0.79–3.7)	0.17
Yes	20 (28%)	52.4 (33.8–81.2)					36.6 (19.9–67.2)				
NLR, NA = 4											
<3	39 (58%)	86.1 (75.5–98.2)	REF 3.0 (1.1–7.6)	0.023	REF 2.6 (1.0–6.8)	0.058	73.6 (60.9–89.1)	REF 1.6 (0.86–3.0)	0.13	NI	NI
≥3	28 (42%)	66.0 (50.0–87.1)					49.2 (33.6–72.0)				

CI: confidence interval; HR: hazard ratio; NA: not available; NI: not indicated; NLR: Neutrophils-to-lymphocytes ratio; OS: overall survival; PFS: progression-free survival; PS: performance status; REF: reference.

**Table 3 biomedicines-10-02758-t003:** Prognostic factors of overall survival and progression-free survival among RCC patients treated with ipilimumab plus nivolumab according to Cox model.

Patient Characteristics	Cox Proportional Hazards Regression for Survival										
		Overall Survival					Progression-Free Survival				
		12 Months OS, % (95% CI)	Unadjusted Analysis		Adjusted Analysis		6 Months PFS, % (95%CI)	Unadjusted Analysis		Adjusted Analysis	
	N (%)		HR (95%CI)	*p*	HR (95%CI)	*p*		HR (95%CI)	*p*	HR (95%CI)	*p*
Age NA = 0											
<70	25 (69%)	80.5 (64.8–100)	REF 1.3 (0.4–4.4)	0.66	NI	NI	63.0 (46.4–85.6)	REF 1.3 (0.5–3.2)	0.55	NI	NI
≥70	11 (31%)	57.7 (35.0–95.0)					62.3 (38.9–99.9)				
PS ≥ 2 (%) NA = 0											
No	32 (89%)	78.2 (63.9–95.7)	REF 7.2 (1.8–29.7)	0.0061	REF 1.8 (0.25–12.8)	0.57	70.8 (56.4–88.9)	REF 6.9 (2.0–23.0)	0.0018	REF 1.5 (0.30–8.0)	0.61
Yes	4 (11%)	NA					NA				
≥3 metastatic site (%) NA = 0											
No	23 (64%)	80.5 (64.8–100)	REF 2.5 (0.8–7.9)	0.12	NI	NI	68.3 (51.3–90.9)	REF 1.4 (0.57–3.4)	0.47	NI	NI
Yes	13 (36%)	57.7 (35.0–95.0)					52.7 (31.2–89.2)				
Bone metastases (%) NA = 0											
No	27 (75%)	82.7 (68.4–100)	REF 3.6 (1.1–11.5)	0.021	REF 1.5 (0.27–8.6)	0.62	77.1 (62.5–95.1)	REF 6.7 (2.4–19.2)	<0.0001	REF 6.4 (1.5–26.8)	0.011
Yes	9 (25%)	41.7 (18.5–94.0)					14.8 (2.6–86.0)				
NLR, NA = 4											
<3	18 (56%)	100 (100–100)	REF 19.4 (2.2–170.7)	0.0075	REF 16.7 (1.8–156.6)	0.014	88.1 (73.9–100)	REF 3.3 (1.2–9.4)	0.026	REF 2.5 (0.79–7.9)	0.12
≥3	14 (44%)	50.0 (27.2–91.9)					48.2 (27.6–84.3)				

CI: confidence interval; HR: hazard ratio; NA: not available; NI: not indicated; NLR: Neutrophils-to-lymphocytes ratio; OS: overall survival; PFS: progression-free survival; PS: performance status; RCC: renal cell carcinoma; REF: reference.

**Table 4 biomedicines-10-02758-t004:** Adverse events in patients with advanced renal cell carcinoma and melanoma treated with ipilimumab plus nivolumab.

Events	Renal Cell Carcinoma (*n* = 36)		Melanoma (*n* = 35)	
	Any	Grade 3–4	Any	Grade 3–4
	number of patients with event (percent)			
Diarrhea	7 (19.4)	1 (2.8)	0	0
Colitis	2 (5.6)	0	2 (5.7)	2 (5.7)
Hepatitis	2 (5.6)	1 (2.8)	13 (37.1)	3 (8.6)
Increase in lipase level	1 (2.8)	1 (2.8)	2 (5.7)	0
Pancreatitis	1 (2.8)	0	0	0
Autoimmune sclerosing cholangitis	2 (5.6)	1 (2.8)	0	0
Pruritus	7 (19.4)	0	14 (40.0)	1 (2.9)
Rash	0	0	15 (42.9)	3 (8.6)
Vitiligo	0	0	20 (57.1)	0
Drug Rash with Eosinophilia and Systemic Symptoms	0	0	1 (2.9)	0
Hypophysitis	2 (5.6)	1 (2.8)	8 (22.9)	2 (5.7)
Diabetes mellitus	0	0	3 (8.6)	0
Thyroiditis	6 (16.7)	0	6 (17.1)	0
Interstitial diffuse lung disease	4 (11.1)	3 (8.3)	0	0
Sarcoidosis	0	0	1 (2.9)	0
Myositis	2 (5.6)	1 (2.8)	1 (2.9)	1 (2.9)
Myocarditis	1 (2.8)	1 (2.8)	0	0
Nephritis	0	0	3 (8.6)	1 (2.9)
Arthralgia	3 (8.3)	2 (5.6)	5 (14.3)	2 (5.7)
Aseptic meningitis	2 (5.6)	2 (5.6)	4 (11.4)	4 (11.4)
Peripheral neuropathy	0	0	3 (8.6)	2 (5.7)
Uveitis	0	0	4 (11.4)	1 (2.9)
Treatment-related adverse event leading to discontinuation	5 (13.9)	5 (13.9)	13 (37.1)	9 (25.7)

## Data Availability

Data available on request due to privacy/ethical restrictions.

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
