# Peer review of "Impact of Bone Metastases on Patients with Renal Cell Carcinoma or Melanoma Treated with Combotherapy Ipilimumab Plus Nivolumab"

_biomedicines, 2022, doi:10.3390/biomedicines10112758_

Round 1

Reviewer 1 Report

Overall, the manuscript is well written, scientifically interesting and logically structured presenting interesting information and novelties with certain statistical limitations.

However, they need to clarify some points.

The number of sample size is low so it seems to be very questionable to make strong enough estimates.

In addition, the authors have not compared survival/progression free survival data from patients with bone metastasis treated with standard therapy with a group of patients treated with standard therapy plus ipilimumab treatment. 

Before accepting the manuscript these clarifications are needed.

Reviewer 2 Report

The study by Pham et al. describes the impact of bone metastases on patients with renal cell carcinoma or melanoma treated with combo therapy ipilimumab plus nivolumab. The authors enrolled the patients in the single-center study. Furthermore, they have retrospectively collected clinical and biological data before initiating ipilimumab plus nivolumab to study the impact of baseline bone metastases on overall and progression-free survival of patients with advanced renal cell carcinoma or melanoma.

Overall, this is a sound study where results are well supported with good data. However, minor issues should be addressed to improve the manuscript's acceptability for publication.

Comments:

1.     What known biomarkers can be used to estimate the risk or benefits of this combined therapy?

2. The authors have mentioned this study's limitations, such as small sample size, single center based, etc. It would be informative to highlight the critical points like "how this study overcomes the limitations of the existing studies. For example, https://www.annalsofoncology.org/article/S0923-7534(22)00083-7/pdf

3. What could be the possible way to overcome these limitations in the future should be included in the concluding sections of the discussion as PERSPECTIVE.
